# Differential Diagnosis between Marfan Syndrome and Loeys–Dietz Syndrome Type 4: A Novel Chromosomal Deletion Covering TGFB2

**DOI:** 10.3390/genes12101462

**Published:** 2021-09-22

**Authors:** Stefano Nistri, Rosina De Cario, Elena Sticchi, Gaia Spaziani, Matteo Della Monica, Sabrina Giglio, Silvia Favilli, Betti Giusti, Pierluigi Stefano, Guglielmina Pepe

**Affiliations:** 1CMSR Veneto Medica-Cardiology Service, 36077 Altavilla Vicentina, Italy; stefanonistri41@gmail.com; 2Department of Experimental and Clinical Medicine, University of Florence, 50134 Florence, Italy; rosina.decario@gmail.com (R.D.C.); elena.sticchi@unifi.it (E.S.); betti.giusti@unifi.it (B.G.); pierluigi.stefano@unifi.it (P.S.); 3Pediatric Cardiology, Azienda Ospedaliera Universitaria Meyer, 50139 Florence, Italy; g.spaziani@meyer.it (G.S.); s.favilli@meyer.it (S.F.); 4Medical Genetics Unit, Cardarelli Hospital, 80131 Napoli, Italy; matteo.dellamonica@aocardarelli.it; 5Unit of Medical Genetics, Department of Medical Sciences and Public Health, University of Cagliari, 09024 Cagliari, Italy; sabrinar.giglio@unica.it; 6Cardiac Surgery Unit, Careggi Hospital, 50134 Florence, Italy; 7Marfan Syndrome and Related Disorders Regional Referral Center, Careggi Hospital, 50134 Florence, Italy

**Keywords:** Loeys–Dietz syndrome, Marfan syndrome, hereditary thoracic aortic aneurysms and dissections, LDS type 4, FBN1, TGFB2, TGFB3, SMAD3, TGFBR1, TGFBR2, chromosomal microdeletion

## Abstract

Marfan syndrome (MFS) and Loeys–Dietz syndrome type 4 (LDS4) are two hereditary connective tissue disorders. MFS displays ectopia lentis as a distinguishing, characterising feature, and thoracic aortic ectasia, aneurysm, dissection, and systemic features as manifestations overlapping with LDS4. LDS4 is characterised by the presence of hypertelorism, cleft palate and/or bifid uvula, with possible ectasia or aneurysms in other arteries. The variable age of onset of clinical manifestations makes clinical diagnosis more difficult. In this study, we report the case of a patient with Marfan syndrome diagnosed at our centre at the age of 33 on the basis of typical clinical manifestations of this syndrome. At the age of 38, the appearance of ectasia of the left common iliac artery and tortuosity of the iliac arteries suggested the presence of LDS4. Next Generation Sequencing (NGS) analysis, followed by Array-CGH, allowed the detection of a novel chromosomal deletion including the entire TGFB2 gene, confirming not only the clinical suspicion of LDS4, but also the clinical phenotype associated with the haploinsufficiency mechanism, which is, in turn, associated with the deletion of the entire gene. The same mutation was detected in the two young sons. This emblematic case confirms that we must be very careful in the differential diagnosis of these two pathologies, especially before the age of 40, and that, in young subjects suspected to be affected by MFS in particular, we must verify the diagnosis, extending genetic analysis, when necessary, to the search for chromosomal alterations. Recently, ectopia lentis has been reported in a patient with LDS4, confirming the tight overlap between the two syndromes. An accurate revision of the clinical parameters both characterising and overlapping the two pathologies is highly desirable.

## 1. Introduction

Loeys–Dietz syndrome (LDS; MIM#609192) is a rare pleiotropic heritable connective tissue disorder (HCTD) displaying an autosomal dominant transmission. This syndrome was first reported by Drs. Bart Loeys and Harry Dietz in 2005, characterised by the following clinical features: hypertelorism, cleft palate and/or bifid uvula, multiple arterial tortuosity, and ascending aortic aneurysm. Other features involve craniofacial, osteoarticular, musculoskeletal, and cutaneous manifestations. LDS presents several clinical features that overlap with Marfan syndrome (MFS; MIM#154700) [1].

Marfan Syndrome: MFS is a rare multisystemic HCTD with autosomal dominant transmission and the following characterising features: thoracic aortic aneurysm (TAA) at the sinus of Valsalva, and/or thoracic aortic dissection (TAD), ectopia lentis, systemic features (SF) with a score indicating the presence of such manifestations =/> 7 to be positive for the clinical diagnosis, in addition to a positive family history (FH) among first relatives and the detection of a pathogenic fibrillin 1 gene (FBN1) mutation already described in patients with thoracic aorta aneurysm/dissection (TAAD) [2].

The clinical diagnosis of MFS is made in the presence of two out of three clinical characteristics: EL, TAAD, SF. Alternatively, we can carry out diagnoses of MFS when one of the three clinical manifestations is present together with the presence of a positive FH or a pathogenic FBN1 mutation [2].

Loeys–Dietz Syndromes: LDS was found to be associated with mutations in TGFBR1 and TGFBR2 genes (encoding Transforming Growth Factor Receptor 1 and 2, respectively) which increase TGF-β (Transforming Growth Factor β) signalling activity [1]. In 2006, Loeys and co-workers subdivided LDSs, also belonging to the syndromic aneurysm disorders, into type 1 (described above) and type 2, characterised by having only bifid uvula in some patients and at least two of the following clinical features of vascular Ehlers–Danlos syndrome (vEDS; OMIM#130050): visceral rupture, wide and atrophic scars, easy bruising, velvety and/or translucent skin, and joint laxity. Overall, LDS type 1 appears to be more severe in terms of untimely death and major complications during pregnancy or in the first postpartum period [3]. At present, six types of LDS and some gene encoding components of TGF-β signalling are known [4]: type 1/TGFBR1 (MIM#190181) and type 2/TGFBR2 (MIM#190182) are still the most severe [1], followed by type 3/SMAD3 (Small Mother Against Decapentaplegic MIM#603109) [5], type 4/TGFB2 (MIM#190220) [6,7], type 5/TGFB3 (MIM#190230) [8], and type 6/SMAD2 (MIM#601366) [9]. Therefore, different components of the TGF-β signalling pathway: cytokines (TGFB2 and 3), receptors (TGFBR1 and 2), and downstream effectors (SMAD2 and 3), mainly displaying a mutation mechanism of loss of function leading to an increase in TGF-β signalling pathway inside the thoracic aortic wall, cause LDS syndromes [9]. FBN1 is a part of the protein complexes which protect and keep TGF-β inactive until it has to activate the pathway; mutations affecting fibrillin 1 concur to hyperactivate TGF-β signalling as it happens for LDSs.

Loeys–Dietz type 4: LDS type 4/TGFB2 is the most similar to Marfan syndrome [6,7]. Both Boileau and Lindsay groups detected the presence of elastin fragmentation and collagen and proteoglycan deposition in the aortic tissue of LDS4 patients, similar to what is found in MFS and other LDSs [6,7]. In LDS4 (MIM#614816), unlike MFS, aortic aneurysm involves mainly, but not exclusively, the sinus of Valsalva at a later onset than the age of thirtyfive [2]. The aortic ectasia is relatively mild and has a lower incidence of dissection, compared to the other LDS types [6]. Among the eight families carrying TGFB2 mutations, Lindsay and co-workers detected two groups of clinical features; the first group common to MFS and LDS including aortic aneurysm, pectus deformity, scoliosis, arachnodactyly, and skin striae. The second group, shared only with LDSs, includes arterial tortuosity, a bicuspid aortic valve, hypertelorism, bifid uvula, thin skin with easy bruising, and club feet. Ectopia lentis was not observed [7].

Chromosomal alterations involving TGFB2: Microarray analysis of two patients who also had mild developmental delay revealed two heterozygous de novo chromosomal microdeletions at 1q41 (Families 1 and 2). The deletion in one proband measures 6.5 Mb (215.5–222.1 Mb; GRCh37/hg19) and encompasses 20 genes, whereas the deletion in the other is only 3.5 Mb (216.6–220.2 Mb). Both deletions include the TGFB2 gene, which encodes the transforming growth factor β2 (TGFβ2). Mutations in TGFB2 display haploinsufficiency as the most relevant mechanism [7]. Other chromosomal deletions were reported by Fontana et al., 5.2 Mb deletion [10]; Gaspar et al., 4.7 Mb deletion [11]; Schepers et al., 3.2 Mb deletion [12].

In this study, we report on a family with a chromosomal deletion involving TGFB2 detected in a father and in his two sons. A literature review of TGFB2 deletions and some observations on the clinical phenotype of the propositus displaying a clear Marfan phenotype until the late onset of other clinical manifestations led us to the suspected LDS type 4.

## 2. Materials and Methods

### 2.1. Patients

Two adult Italian brothers were evaluated at the Marfan and Related Disorders Tuscany Regional Referring Center, while, some years later, the two sons of one of the brothers were evaluated at the Meyer Children’s Hospital of Florence, Italy.

### 2.2. Imaging Analysis

Aortic dimensions were assessed by a trans-thoracic echocardiography at end-diastole in the parasternal long-axis view at the sinuses of Valsalva, the sinotubular junction, and the proximal ascending aorta by the leading edge-to-leading edge technique. The Z-score was calculated according to age-adjusted nomograms [13,14].

### 2.3. Genomic DNA Preparation

Peripheral venous blood was collected in EDTA Vacutainer tubes and stored at −20 °C. Genomic DNA was extracted from blood samples using a FlexiGene Kit (Qiagen, Hilden, Germany) according to the manufacturer’s instructions.

### 2.4. Next Generation Sequencing (NGS) Analysis

A targeted NGS was performed on a 97-gene panel applied in the diagnostic and research routine for Marfan syndrome and related disorders. Among the genes, we selected the following: FBN1, TGFBR1, TGFBR2, TGFB2, TGFB3, and SMAD3 to be analysed. Oligo-probes specific for the target gene regions were designed using Agilent Sure Design (Agilent Technologies, Santa Clara, CA, USA) in order to create a custom target-enriched library of the selected genes. The capture region comprised all coding exons and flanking intron sequences (50 bp upstream and downstream at exon–intron junctions). The final design consisted of a cumulative targeted region of 627.782 kb. Amplicon sequencing libraries were prepared from 500 to 750 ng/μL of DNA per sample according to the SureSelectQXT Amplicon protocol (Illumina Inc., San Diego, CA, USA). The pooled libraries were paired-end and sequenced on a micro-flow cell with V3 chemistry on a MiSeq instrument (Illumina Inc., San Diego, CA, USA). The analytical pipeline available in our laboratory was developed, implemented, and validated for data analysis of targeted sequencing for diagnostic purposes [15].

### 2.5. Alignment and Variants Calling

Fastq files’ quality was checked with FASTQC. Adapters and quality trimming were performed using Surecall Trimmer. Trimmed reads were aligned to the human reference genome (Human GRCh37/hg19) using BWA-MEM. Bam files’ quality was evaluated with Qualimap. Variant calling was performed using GATK4 HaplotypeCaller in GVCF mode and the joint genotyping tool GenotypeGVCFs. Variants were annotated using VEP 99 [15].

### 2.6. Array-CGH Analyses

Array-CGH analyses were performed at the clinical genetic laboratory of the Meyer Children’s Hospital (Florence, Italy) [16].

## 3. Results

### 3.1. Patient and His Family

The propositus is a 43-year-old male (Figure 1, proband II-2), who was diagnosed with MFS at 33 years old. The patient displayed an aortic root of 47 mm (z-score 4.4), with normal abdominal aortic size, systemic features with a score of 8, and hypermobility Beighton scale score of 8/9 (Table 2). The diameter of the abdominal aorta was normal. After 2 and 5 years, the aortic root diameter slowly progressed to 48 and 49 mm (z-score 4.6), respectively. A mild abdominal ectasia (32 mm) was detected at the age of 38. In the meantime, a second son was born; both were not referred to our centre despite our suggestion.

At 40 years old, Computed Tomographic angiography (CTA) of the entire aorta confirmed the ectasia of the aortic root (Figure 2) and revealed mild ectasia of the common left iliac artery (16 × 14 mm) with postostial kinking and tortuosity of the iliac arteries (Figure 2). We have seen both MRI and angio-TC images in 2018 for the first time.

At the age of 41, the patient underwent valve sparing aortic root replacement (David procedure) for a 50 mm root aneurysm. Importantly, the patient reported that his father had hypertension and his mother pectus carinatum, pes planus, and inguinal hernias. Echocardiography did not show cardiovascular features. The parents at present (mother is 74 and father is 76 years old) are alive and in good health with regard to MFS, which is why eye clinical ocular manifestations were excluded by the ophtalmologist. The older brother (age 47) has a “potential” Marfan phenotype [2], stable over time, that was diagnosed at our centre. The youngest son of the propositus (Figure 1, III-2) was hospitalised in a neonatal intensive care unit due to a supraventricular arrhythmia with hemodynamic impairment diagnosed at birth (Figure 3); the atrial flutter was successfully treated with DC-shock (DC = direct current) and the clinical status of the neonate quickly improved.

The older son of the propositus (Figure 1, III-1), assessed by the Meyer geneticists and cardiologists two years ago when he was 6 years old (Figure 4), displayed hypermobility and fetal valgus feet, which were also detected in the younger son (Figure 1, III-2) at the age of 3. The echocardiograms of both sons confirmed the absence of structural heart disease during follow-up; in particular, the image showed normal size of the aortic root (Figure 4) in the younger son (Figure 1, III-2) at 3 years old, as well as an annulus of 16 mm (z score + 1.33), Valsalva sinus of 21 mm (z score + 0.91), sino-tubular junction of 18 mm (z score + 1.31) and an ascending aorta of 17 mm (z score + 0.65). Additionally, the older son (Figure 1, III-1) presented a normal value of aortic root at 6 years old, as well as an annulus of 16 mm (z score + 0.24), Valsalva sinus of 24 mm (z score + 1.11), sino-tubular junction of 18 mm (z score − 0.96), and an ascending aorta of 17 mm (z score − 0.33).

### 3.2. Genetic Analysis

NGS analysis was performed on the father (Figure 1, II2), but he turned out to be negative for FBN1 and the other genes analysed. Additionally, the brother of the propositus (Figure 1, II3), had a negative result from the analysis of the FBN1 gene. An array-CGH analysis revealed an interstitial deletion of 0.248 Mbp+/− 0.050 Mbp in chromosome 1 (1q41). The deletion here reported included the TGFB2 gene. According to the genetic medical report, no patients were reported as having the same deletion in the University of California Santa Cruz (UCSC) database. The analysis, extended to the family, revealed the presence of the deletion in both sons (Figure 1, III-1 and III-2). The brother (Figure 1, II-3) has not yet undergone cytogenetic analysis. Unsuccessfully, genetic analyses were proposed many times to the propositus’ parents and brother.

## 4. Discussion

Our patient (Figure 1, II-2) presented the typical manifestations of MFS (Table 1) up to the age of 38: root thoracic aorta ectasia and systemic features, among which he had tall stature, arachnodactyly of upper and lower limbs, positive wrist and thumb signs with a score of 8, and hypermobility with a Beighton score of 8/9. Hypertelorism, cleft palate, bifid uvula, osteoarthritis, and ectopia lentis were absent. The clinical diagnosis was reinforced by the patient’s brother (Figure 1, II-3) with potential Marfan syndrome. No other particular ocular or cardiovascular manifestations were detected in either (Table 1).

The onset of abdominal ectasia in the left common iliac artery with kinking suggested the new diagnosis of LDS, probably type 4, the most similar to Marfan syndrome. Genetic analysis was previously performed on candidate genes for MFS and LDSs, but the result was negative. Later, array-CGH analysis revealed the presence of a chromosomal deletion of approximately 248 Mbp containing the TGFB2 gene, confirming the suspected clinical diagnosis of LDS4. This result changed the clinical diagnosis of the propositus (Figure 1, II-2), which is now classified as affected by Loeys–Dietz syndrome type 4. Array-CGH analysis was also performed on the two sons (Figure 1, III-1, III-2), who both carry the deletion, while the brother (Figure 1, II-3) has not been blood tested yet. Deletions of the entire TGFB2 gene cover less than 15% of TGFB2 mutations at present and, together with nonsense mutations, splicing defects and frameshift mutations represent the most common disease mechanism of this gene: haploinsufficiency. The deletions involving the entire TGFB2 gene are of all different sizes (Figure 5). From the largest to the smallest, they present the following sizes in Mb: 6.5 [7], 5.2 [10], 4.7 [11], 3.2 [12], 3.5 [7], and 0.25 (this study).

Each deletion is different in size, from the largest of 6.5 Mb (Lindsay 2012, shown in red), followed by that of 5.2 Mb (Fontana 2014, in green), that of 4.7 Mb (Gaspar 2017, in light blue), that of 3.5 Mb (Lindsay 2012, in green bean), and that of 3.2 Kb (Schepers 2017, in gray), to the smallest of 0.25 Mb (reported in this study, shown in cyan blue). The last one causes the deletion of only one other gene, RRP15, in addition to TGFB2. The 0.25 Mb deletion corresponds to the most representative clinical phenotype of the LDS4 being involved in the deletion with only one other gene that should be recessive.

The chromosomal deletion reported in this study eliminates the entire TGFB2 gene as well as another gene, RRP15 (MIM # 611193), which has unknown functions in humans. RRP15 in Saccharomyces cerevisiae is a component of the early pre-60 ribosomal particles, which are required for the maturation of large subunit ribosomal RNA. The deletion shown in this report is very important because it confirms that the isolated deletion of a TGFB2 allele without the interference of many other deleted genes around it (Table 2) responds to the clinical features of our patient, which are due to the loss of one copy of the TGFB2 gene. While our patient has a chromosomal deletion involving only two genes, one of which has no known function in humans, the other patients with LDS4 reported in Table 2 have a much larger deletion that causes the loss of many genes, including some with known functions and others with unknown functions. Interestingly, the patients display different clinical phenotypes, and among this small group of LDS4/TGFB2 patients, our patient is the only one displaying aortic/arteries ectasia or aneurysm, while one patient had type B aortic dissection (Table 2 This result changed the clinical diagnosis of the propositus (Figure 1, II-2). In general, the variable sizes of the deletions correspond to a variable area and the number of genes involved together despite different ages and sexes of the patients may, at least in part, respond to the variable phenotype. Moreover, the patient described by Fontana and co-workers [10] has four different chromosomal deletions, causing a complicated phenotype. The deleted genes encoding proteins the most, although displaying an autosomal recessive expression, may act as modifiers of the final phenotype (Table 2). We can compare the clinical data of patients based on their age, with the patients reported in the first, fourth, and sixth (this report) columns (Table 2), aged between 40 and 46 years, and the others whose ages are 9, 12 and 18 years, respectively, are probably too young to display the cardiovascular features and all the other LDS4 manifestations (Table 2). How can we explain the absence of vascular features? Since isolated point mutations in TGFB2 cause the clinical manifestations reported in Table 2 (column 5), we can hypothesize that, among the many genes lost in the other patients with larger chromosomal deletions, one or more genes may act as positive modifier genes against the onset of aortic ectasia/aneurysm or the loss of chromosomal areas containing regulator genes may inhibit the onset of aortic ectasia. No other data have been reported on the vascular system of the five patients.

Moreover, the brother of the propositus (Figure 1, II-3) has been diagnosed as “potential” Marfan, a clinical pattern common among the Marfan sufferer’s relatives, who was clinically stable at the age of 47. Instead, at the age of 40, an angio-TC revealed in the propositus (Figure 1, II-2) the onset of mild ectasia of the left common iliac abdominal aorta and tortuosity of the iliac arteries, which required a differential diagnosis between MFS and LDS.

Since not all patients in differential diagnoses between MFS and LDS4, turning out to be negative for NGS analysis, undergo array-CGH analysis, this family report shows that it would be appropriate in the clinical follow-up to perform periodic checks of the epiaortic vessels and of the aorta as a whole since the differential diagnostic doubt is not yet completely resolved. It is also important to underline the fact that, recently, an article has been published describing an LDS4 patient with ectopia lentis [17]. This is of little significance at the moment, with it being a single and sporadic case, but it is still a further alarm bell for a correct differential diagnosis between MFS and LDS4 and, above all, for an appropriate clinical follow-up and surgical timing.

## 5. Conclusions

The description of this new familiar case of LDS4, with deletion of the entire TGFB2 gene and involving only one more gene, allows us to verify the clinical phenotype related to the loss of one entire copy of the gene. Moreover, it suggests that in the 10% of MFS cases in which we do not find a FBN1 mutation, we have to search for gene and chromosomal deletion/insertions with the aim of performing a differential diagnosis between MFS and LDS4 with immediate consequences for appropriate clinical follow-up and surgical timing of the patients.

## Figures and Tables

**Figure 1 genes-12-01462-f001:**
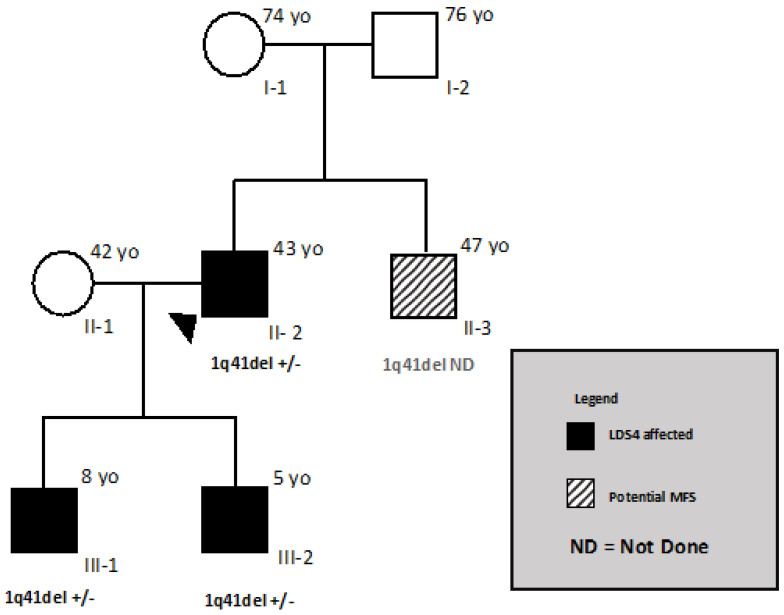
Chromosomal microdeletion encompassing TGFB2 gene causes LDS type 4.

**Figure 2 genes-12-01462-f002:**
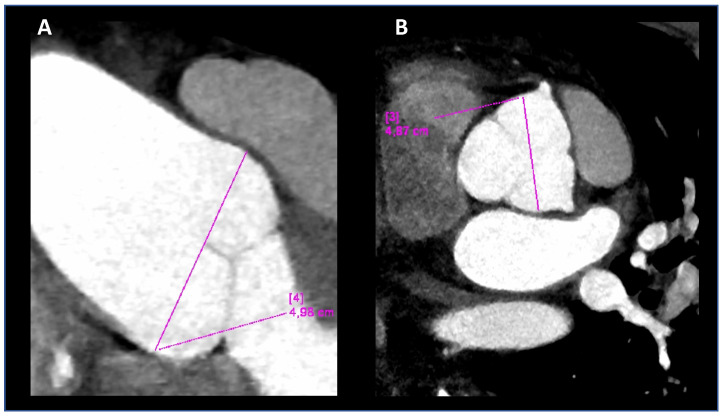
The computed tomographic angiography (CTA) of the thoracic aorta, performed on the propos-itus at 40 years. Coronal oblique (**A**) and double oblique (**B**) images, allowing measure-ments in long- and short-axis views, respectively. The aortic root diameters at Valsalva sinuses were 48 and 49 mm.

**Figure 3 genes-12-01462-f003:**
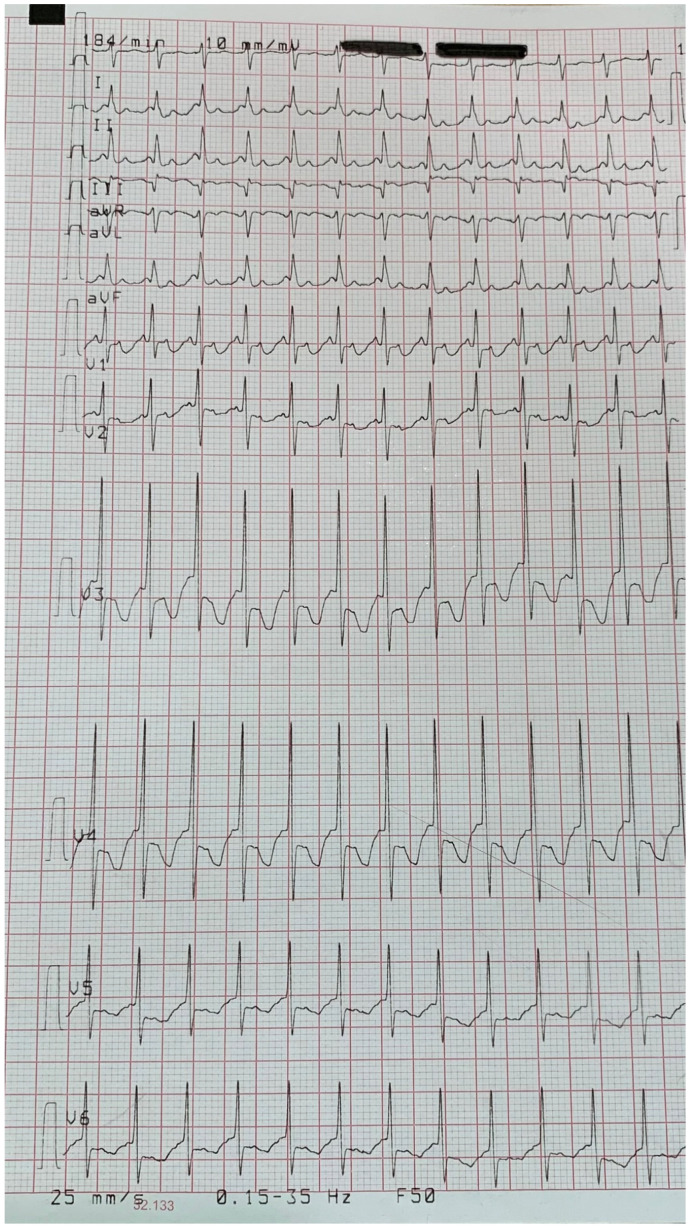
The electrocardiogram of the youngest son of the propositus (Figure 1, III-2) revealed an atrial flutter at birth which required hospitalisation and DC-shock therapy.

**Figure 4 genes-12-01462-f004:**
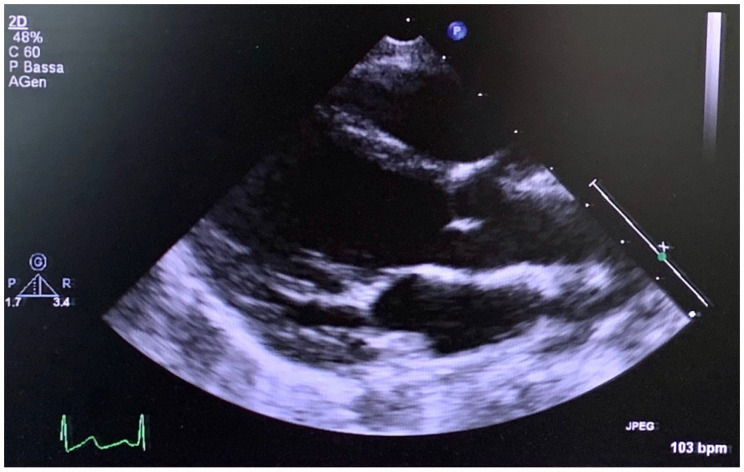
At 3 years old, the echocardiogram (long axis parasternal view) of the youngest son of the propositus showed both a normal size of the aortic root and the left ventricle.

**Figure 5 genes-12-01462-f005:**
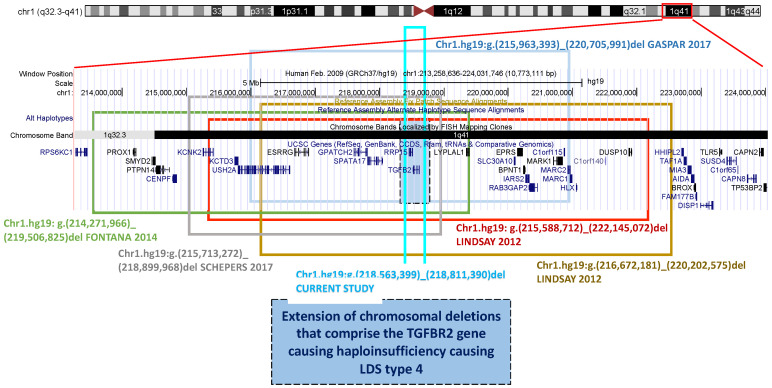
Schematic representation of the 6 1q41 deletions, including the one presented in this study. Each colored rectangle marks a single chromosomal deletion associated with a patient with LDS4.

**Table 1 genes-12-01462-t001:** MFS = Marfan syndrome; LDS4 = Loeys–Dietz type 4; FBN1 = fibrillin1 gene; TGFB2 = Transforming Growth Factor Beta2; SF = systemic features; US = upper segment (of the body); LS = lower segment (of the body); Ds = diopters; CVS = cardiovascular; TAD = thoracic aorta dissection; N.A. = not analysed; columns 4 and 5: “-” = absent or at population frequency; “+” = some cases reported; “++” = common in this disease; “+++” = frequent in this disese [6,7,12]; I don’t have a copyright but I don’t think I have to ask for that because these results are the sum of results from more articles. I will change the definitions and mention all the references columns 6 and 7: “+” = present; “-” ok for “--” = absent. (1), (2), (3) = characteristic clinical features of MFS.

Clinical Manifestations			MFS	LDS4	Figure 1, This Report	
	Genes		FBN1	TGFB2	II-2	II-3
Systemic Features (1) Score =/> 7	Score of SF	Systemic Features	Clinical Features	Clinical Features	Score of SF	
Facial Features:					1	1
	1	dolichocephaly			--	+
If present 3/5 features		downslanting palpebral fissures			+	--
		enophthalmos			+	+
		malar hypoplasia			+	+
		retrognatia			+	+
Body segments	1	Reduced US/LS AND increased arm span (AS)/height (H) AND no severe scoliosis			1 (US/LS0.54, AS/H = 1.06)	--
					+	+
Pectus deformity						
	2	carinatum	++	++	2	--
	1	Excavatum or chest asimmetry			--	1
Rachis	1	>20 °C Scoliosis or thoracolumbar kyphosis			+	--
Upper limb	1	Reduced elbow extension			--	1
	3	Wrist AND thumb sign			3	--
	1	(wrist OR thumb sign				
	2	Protrusio acetabuli			N.A.	N.A.
Lower limb	2	Hindfoot deformity			--	2
	1	plain pes planus			1	1
	2	Dural ectasia (DE)	+	+	--	N.A.
	2	Pneumotorax (PNX)			--	--
	1	Mitral Valve Prolapse (MVP, any type)			--	--
	1	Myopia >3Ds			+	+
	1	Skin striae	++	+	1	1
CVS		Aortic root aneurysms (2)	++	++	+	--
		TAD (2)	+	+	--	--
Other CVS		Ascending aorta aneurysm			--	--
		Other aneurysms	+	+	+ late onset	--
		Arterial tortuosity	-	+	+ late onset	--
		BAV(bicuspid aortic valve)	+	++	--	--
(C) Eyes		Ectopia lentis (EL) (3)	+++	--	--	--
		Cleft palalate/bifid uvula	--	+	--	--
		Hypertelorysm	--	+	--	--
		Tall stature	+++	++	+	+
		Arachnodactyly	+++	+	+ hands and feet	+
		Clubfoot	--	++	--	--
		Osteoarthritis	++	+	--	--
		Hernia	+	+	+	--
		Hypermobility			+	--
GENES		FBN1			-	--
		TGFB2arrayCGH			+	N.A
		TGFB2 NGS			-	N.A.

**Table 2 genes-12-01462-t002:** Clinical manifestations of each patient carrying TGFB2 deletion from the literature. The one from Schepers’ paper was not added because clinical data were not available. The clinical manifestations of MFS, present in the first outpatient visit (patient reported in this report) are highlighted in light blue, while those typical of LDS appearing/detected at the age of 38 are highlighted in green. Mb = megabases, M = male, F = female, “-” = absent, “+” = present, N.R. = not reported, TAV = tricuspid aortic valve, MVP = mitral valve prolapse.

Reference and PatientID	Lindsay 2012 II-1	Lindsay 2012 II-2	Fontana 2014	Gaspar 2017 II-2	Gaspar 2017 III-1	This Report
Mutation	1pq41 ch del	1pq41 ch del	1pq41 ch del	1pq41 ch del	1pq41 ch del	1pq41 ch del
Deletion size	6.5 Mb	3.5 Mb	5.2 Mb	4.7 Mb	4.7 Mb	0.25 Mb
Number of deleted genes known to encode proteins	20	9	15	18	18	2
Sex	M	M	F	F	M	M
Age	46	9	18	40	12	43
Craniofacial						
Eye	myopia	hyperopia	Severe myopia, strabismus, exotropia, ptosis, nystagmus progressive tapeto-retinal degeneration, blue sclera	-	-	myopia
Downslant palpebral fissures	-	+	+	+	+	+
Hypertelorism	+	-	+	-	-	-
High arched palate	+	+	+	-	-	+
Uvula	N.R.	N.R.	-	-	-	-
Retrognathia	+	+	+	N.R.	+	+
OTHER	torticollis	ptosis	Triangular face, low-set ears, thin lips, mild conductive hypoacusis, dental enamel hypoplasia		Dental enamel hypoplasia, abn, anteversal nares	
Skeletal						
Stature cm or percentile	193	195	145.7	154		200
Armspan ratio	1.02	0.96	N.R.	N.R.		1.04
Pectus Deformity	+	+	+	N.R.	+	+
Scoliosis	+	-	+ Dorso-lumbar scoliosis	N.R.	N.R.	+
Arachnodactily	+	+	N.R.	+	-	+
Positive thumb/wrist	-	-	-	+	-	+
Club feet	+	+	-	-	N.R.	-
Pes planus	-	+	-	N.R.	N.R.	-
Joint Hypermobility > 5/9	N.R.	N.R.	N.R.	+ (7/9)	+ (6/9)	+ (8/9)
OTHER			Mild motor, language delay, optic canal hyperostosis, coxa vara, genu varu, coxa valga surgery, delta-storage pool pt disease	Osteoporosis bilateral femoral and neck, fractures of the pelvis and the ribs, muscle weakness, chronic pain	Muscular hypotonia, problems with motor coordination, dyslalia	
Cardiovascular						
Aortc root-z-score	2.8	3.0	2.35	N.R.	-	5.0 before surgery
Ao	-	-	31 normal size	Normal size	Normal size	Surgery at 38 years
Ao dissection/repair	typeB, age42	-	-	-	-	
Aortic valve	TAV	TAV	TAV	TAV	TAV	TAV
Mitral valve	MVP	-	- Redundant cusps	-	-	-
Arterial aneurysm	-	-	-	-	-	Left common iliac, abdominal aorta, mild ectasia 14 × 16 (mm)
Arterial tortuosity	-	-	-	-	-	Arterial tortuosity iliacs
OTHER	Claw toes	Claw toes				
Skin						
Striae	-	-	-	-	-	+
Hernia	+	-	-	-	+ inguinal right surgery	+
Easy bruising	-	-	+	-	+	-
OTHER			Hematomas, nose bleeding			
dura	ND	ND	ND	ND	ND	Periradicular cysts
Other findings	cryptorchidism	Hypotonia, ataxia	Epileptic seizures	Excluded learning disability	hypothyroidism	Varicose veins, deep venous thrombosis

## Data Availability

Not applicable.

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
