# Peer review of "Differential Diagnosis between Marfan Syndrome and Loeys–Dietz Syndrome Type 4: A Novel Chromosomal Deletion Covering TGFB2"

_genes, 2021, doi:10.3390/genes12101462_

Round 1

Reviewer 1 Report

This study reported a case of Loeys-Dietz syndrome type 4 (LDS4) with deletion of the entire TGFB2. The patient met the diagnosis of Marfan syndrome (MFS) based on the 2010 Ghent criteria but had no typical manifestations (Hypertelorism, bifid uvula) of LDS. Therefore, the authors tried to propose that a revised clinical-genetic diagnostic path is highly desirable.

However, the situation in this case is already included in the Ghent criteria. The statement of revision for the current criteria may exaggerate the finding. It would be more appropriate to focus on the deletion mutation in the absence of identical mutations in MFS and LDS. In addition, genetic analysis and medical examinations of the whole family members need to be supplied.

Major comments:

1) Since the propositus mother had some systemic features, please include more details such as the cardiovascular manifestations, age (survival or not), possible genetic analysis.

2) The propositus brother phenotype was described as “potential Marfan Syndrome”. This is vague. The authors need to provide specific information for the potential MFS diagnosis based on Revised Ghent criteria (2010). Although, the authors mentioned the brother did not have a genetic test, it should be more rigorous and complete with genetic analysis.

3) Based on the phenotype provided by the authors (z-score 4.4, systemic score 8), the propositus fulfilled the “In the absence of family history: Ao (Z ≥2) AND Syst (7≥ pts) =MFS*” in 2010 Revised Ghent criteria. It should be noticed that the 2010 Ghent criteria had already pointed out “*: without discriminating features of SGS, LDS or vEDS AND after TGFBR1/2, collagen biochemistry, COL3A1 testing if indicated. Other conditions/genes will emerge with time” in this situation. In this report, the subsequent detection of TGFB2 mutation and diagnosed as LDS4 met the condition outlined above in 2010 Ghent criteria. In conclusion, the revised Ghent nosology is still suitable for this case, it might be excessive or meaningless to raise the concept of “revise the current diagnostic path” based on this report.

4) Arterial tortuosity is a distinguishing feature in LDS compared with MFS in the majority of cases. This feature should be noticed when the patient under CT angiography (CTA) of aorta at age of 40 or earlier. In the absence of FBN1 mutation and ectopia lentis and with arterial tortuosity, it should had caused the attention of the authors that the patient might have LDS.

5) From a clinical perspective, treatment and the time of intervention for aortic aneurysm in LDS4 seem to be similar to MFS. The differential diagnosis of these two diseases is important. However, no convincing explanations are obtained from this report. A more in-depth discussion of the “an appropriate clinical follow-up and surgical timing” mentioned in the Conclusion part (Page 8, Line 229) should be provided.

Minor comments:

1) Ethical approval: are there projects and approval numbers by the ethical committees that could be presented?

2) The first paragraph of the Introduction is much too long. It could be split into several paragraphs.

3) It may be clearer to call the “the Valsalva level” (Page 2, Line 53) by “the sinus of Valsalva” or “aortic root level”.

4) In multiple places (Page3, Line 140; Page 4, Line 163), the word “carinatum” was spelled wrongly as “carenatum”.

5) All the objects within Figure 1 need to be clearly labeled including the white square.

6) Table and its legend (Page5, Line 192-194) require numbering and careful rearranging.

7) Please provide some cardiac imaging data (CTA of aorta, echocardiography) and pictures of facial or skeletal features of the family members (especially the propositus). These data would make the report more intuitive.

Author Response

Dear Reviewer,

Thank you for your comments, suggestions and criticisms that have surely improved the quality of the article.

Below the answers to your comments

This study reported a case of Loeys-Dietz syndrome type 4 (LDS4) with deletion of the entire TGFB2. The patient met the diagnosis of Marfan syndrome (MFS) based on the 2010 Ghent criteria but had no typical manifestations (Hypertelorism, bifid uvula) of LDS. Therefore, the authors tried to propose that a revised clinical-genetic diagnostic path is highly desirable.

However, the situation in this case is already included in the Ghent criteria. The statement of revision for the current criteria may exaggerate the finding. It would be more appropriate to focus on the deletion mutation in the absence of identical mutations in MFS and LDS. In addition, genetic analysis and medical examinations of the whole family members need to be supplied.

Following the above comments we edited results and discussion. We also changed the title of the article.

Major comments:

  • Since the propositus mother had some systemic features, please include more details such as the cardiovascular manifestations, age (survival or not), possible genetic analysis.

According to your requests we added information about the mother .lines   182-185    

Many times we proposed to the all family (parents and brother) genetic analyses but they haven’t accepted yet. lines 220-221

2) The propositus brother phenotype was described as “potential Marfan Syndrome”. This is vague. The authors need to provide specific information for the potential MFS diagnosis based on Revised Ghent criteria (2010). Although, the authors mentioned the brother did not have a genetic test, it should be more rigorous and complete with genetic analysis.

A new Table, Table 1, (lines between 166 and 167) has been added with all the clinical data of the two brothers. As required by another reviewer the Table presents also the clinical manifestations required for the diagnosis of both Marfan syndrome and Loeys-Dietz type 4 syndrome. We also report the exclusion of mutations in FBN1 by NGS analysis of the propositus and a direct sequencing analysis of the brother requested to the hospital lab center as diagnostic analysis (Table 1, lines 214-215).

3)Based on the phenotype provided by the authors (z-score 4.4, systemic score 8), the propositus fulfilled the “In the absence of family history: Ao (Z ≥2) AND Syst (7≥ pts) =MFS*” in 2010 Revised Ghent criteria. It should be noticed that the 2010 Ghent criteria had already pointed out “*: without discriminating features of SGS, LDS or vEDS AND after TGFBR1/2, collagen biochemistry, COL3A1 testing if indicated. Other conditions/genes will emerge with time” in this situation. In this report, the subsequent detection of TGFB2 mutation and diagnosed as LDS4 met the condition outlined above in 2010 Ghent criteria. In conclusion, the revised Ghent nosology is still suitable for this case, it might be excessive or meaningless to raise the concept of “revise the current diagnostic path” based on this report.

According to your comments the discussion had been changed (254-256, 266-272, 285-291) as well as the title of the manuscript. (lines 3-5)

  • Arterial tortuosity is a distinguishing feature in LDS compared with MFS in the majority of cases. This feature should be noticed when the patient under CT angiography (CTA) of aorta at age of 40 or earlier. In the absence of FBN1 mutation and ectopia lentis and with arterial tortuosity, it should had caused the attention of the authors that the patient might have LDS.

Yes, I agree with the reviewer. We suspected LDS4 when we read the CT report on artery ectasia and artery tortuosity.(lines 170-173)

5) From a clinical perspective, treatment and the time of intervention for aortic aneurysm in LDS4 seem to be similar to MFS. The differential diagnosis of these two diseases is important. However, no convincing explanations are obtained from this report. A more in-depth discussion of the “an appropriate clinical follow-up and surgical timing” mentioned in the Conclusion part (Page 8, Line 229) should be provided.

According to your comments the discussion and conclusion have been changed (254-256, 266-272, 285-291, 293-298)

Minor comments:

  • Ethical approval: are there projects and approval numbers by the ethical committees that could be presented?

This is required by the genes editorial office, therefore we provided them with the above information

2) The first paragraph of the Introduction is much too long. It could be split into several paragraphs.

Yes, it has been splitted in 4 paragraphs.

3) It may be clearer to call the “the Valsalva level” (Page 2, Line 53) by “the sinus of Valsalva” or “aortic root level”.

The “Valsalva level” has been substituted by “the sinus of Valsalva” according to your request. (lines 55-56)

  • In multiple places (Page3, Line 140; Page 4, Line 163), the word “carinatum” was spelled wrongly as “carenatum”.

Yes, thank you “carenatum” has been substituted by “carinatum”.

5) All the objects within Figure 1 need to be clearly labeled including the white square.

we improved the labelling

6) Table and its legend (Page5, Line 192-194) require numbering and careful rearranging.

Done

7) Please provide some cardiac imaging data (CTA of aorta, echocardiography) and pictures of facial or skeletal features of the family members (especially the propositus). These data would make the report more intuitive.

We showed some echocardiographic picture

Unfortunatly at the moment we have no permission to publish pictures of the family. We added cardiac imaging of the propositus and his sons

Reviewer 2 Report

Thank you for the opportunity to review the manuscript “Differential diagnosis between Marfan syndrome and Loeys-Dietz syndrome. Is it time to revise the clinical-genetic diagnostic path?” by Stefano Nistri et al. Marfan syndrome (MFS) and even more Loeys–Dietz syndrome (LDS) are rare, genetically and clinically heterogeneous disorders. Therefore, I find the paper interesting and valuable for both geneticists and clinicians. 

However, there are some points that need to be improved.

Major comments:

  1. I would recommend explaining in more detail the background of the MFS and LDS, the significance of involved genes.
  2. The deletion found in the patient and his sons is small and as far as I understand has not been reported before. Please explain more clearly the significance of the genetic result and the connection to the clinical picture.
  3. In my opinion the Authors could include a Table regarding the MFS and LDS clinical features with the description of a clinical score. Therefore, the text will be easier to follow (lines 53-54- “a score =/> 7 to be positive for the clinical diagnosis”).
  4. Lines 55, 56, 65 and following- please check and explain the abbreviations (FBN1 mutation, TAAD, TGFBR1&2 genes, TGF-beta signalling activity, type 3/SMAD3, etc.). The abbreviations in Tables and Figures should be explained as well.
  5. I would recommend adding an additional Table or a Figure with clinical features and scores for the presented case and the family members.
  6. English language needs extensive editing.
  7. I think the paper should be rather classified as a case report.

Author Response

Dear Reviewer,

Thank you for your comments, suggestions and criticisms that have surely improved the quality of the article.

Below the answers to your comments

Thank you for the opportunity to review the manuscript “Differential diagnosis between Marfan syndrome and Loeys-Dietz syndrome. Is it time to revise the clinical-genetic diagnostic path?” by Stefano Nistri et al. Marfan syndrome (MFS) and even more Loeys–Dietz syndrome (LDS) are rare, genetically and clinically heterogeneous disorders. Therefore, I find the paper interesting and valuable for both geneticists and clinicians. 

However, there are some points that need to be improved.

Major comments:

  1. I would recommend explaining in more detail the background of the MFS and LDS, the significance of involved genes.

The clinical background of MFS and LDS has been reported in the new Table 2. Comments on the significance of the involved genes have been added (76-79)

  1. The deletion found in the patient and his sons is small and as far as I understand has not been reported before. Please explain more clearly the significance of the genetic result and the connection to the clinical picture

Following your request we added informations and comments in the abstract and in the Discussion (lines 32-33, 254-256, 266-272)

  1. In my opinion the Authors could include a Table regarding the MFS and LDS clinical features with the description of a clinical score. Therefore, the text will be easier to follow (lines 53-54- “a score =/> 7 to be positive for the clinical diagnosis”).

Thanks for your suggestion, we added Table 1 with all the clinical characteristics of Marfan (Loeys 2010) and Loeys-Dietz syndromes (Lindsay 2012). We also outlined the manifestations that allow the diagnosis of both (Loeys 2010 and Lindsay 2012) and the features that are more or less expressed by MFs and LDSs (Linday 2012) (between lines 166-167)

  1. Lines 55, 56, 65 and following- please check and explain the abbreviations (FBN1 mutation, TAAD, TGFBR1&2 genes, TGF-beta signalling activity, type 3/SMAD3, etc.). The abbreviations in Tables and Figures should be explained as well.

Abbreviations have been explained according to your requests (lines 58, 60-62)

  1. I would recommend adding an additional Table or a Figure with clinical features and scores for the presented case and the family members.

According to your request together with the clinical manifestations of MFS and LDS4 we added the clinical manifestations of the propositus and the brother in Table 2. We also added imaging of the aorta of the proband and his son as requested by another reviewer (lines betwee 166-167, 174-175, 205-206, 190-191).

I never saw the two sons, as I wrote in the article,  but I reported the only clinical manifestations they presented described by Dr Della Monica, a medical geneticist and co-author in this article. You know that it is very unprobable to reveal clinical manifestations of MFS or LDS4 in very young children (they were seen at the age of 3 and 6 the last time; they will come back in 2022) and we could exclude the syndrome in the parents of the propositus. (lines 187-205)

  1. English language needs extensive editing.

The article has been revised by a English mother tongue

  1. I think the paper should be rather classified as a case report.

I leave the decision to the Editor

Round 2

Reviewer 2 Report

Please see the attached review.

Author Response

Dear Reviewer,

first of all we have to thank you for your patience in re-reading this article that you judged in the second review "chaotic and difficult to understand in some points"

For this reason we have answered to all your requests and suggestions and in doing so, paying attention to your positive criticism, we have also revised and rearranged, with small corrections that I will report below, the various sections of the work, the figures and the tables necessary after the revision of the discussion.

I hope you can now find the article more clear.

  1. Lines 158-167  -  in my opinion this information should be  in the main text, not the Table 1 title.

Following your request, the information has been removed from Table 1 title   and inserted in the text: lines 61-64; 97-101

  1. Table 1 – still not all abbreviations are explained

All abbreviations of Table 1 have been explained

      In red all additions are indicated

  1. The manuscript seems to be chaotic

Line 145 “the propositus is a 43-year-old male (Fig. 1, proband II,2) who at 33 was with Marfan” – was suspected? Diagnosed?

The sentence has been changed in

 “the propositus is a 43-year-old male (Figure 1, proband II,2) who at 33 was diagnosed with MFS” Lines 148-149 

Line 189-190  “Echocardiography did’nt show neither eye problems nor cardiovascular features” – please revise

The sentence has been changed in

“Echocardiography did not show cardiovascular features of MFS, while eye clinical ocular manifestations were excluded by the ophtalmologist.” 

Line 234  “Unsuccessfully, (in vain) analyses were proposed many times to the parents and brother.”   

The sentence has been changed in

 “Unsuccessfully, genetic analyses were proposed many times to the propositus’ brother.” Lines 229   

Lines 237-240 – please, make the text more understandable

The sentence has been changed in

“Our patient (Figure 1, II-2) presented the typical manifestations of MFS (Table 1) up to the age of 38: root thoracic aorta ectasia, systemic features, among which tall stature, arachnodactyly of upper and lower limbs, positive wrist and thumb sign with a score of 8, hypermobility with a Beighton score of 8/9.” Lines 230-233

Lines 267-265 – please make the text moru understandable; in my opinion thys information should be in the main text, not the figure 5 title    and

Lines 267-271 – please make the text more understandable

To make part of the discussion more understandable we re-wrote part of the discussion.    Lines 262 -294

  Line 280 – “column with an age between 40 and 46, the other whose age is 9, 12 and 18, respectively” – “years” should be added, Line 282 – “years” has been added to both columns Line 295 – lack of Figure number and description?

The text of Figure 5 has been changed:

“Figure 5: Schematic representation of the 6 1q41 deletions including the one presented in this study. each colored rectangle marks a single chromosomal deletion associated with a patient with LDS4” lines 251-252 

We double-checked all the numbers of the figures and tables, the descriptions and the abbreviations

Lines 318-324. the conclusions have also been slightly changed  We have reinserted in the bibliography the numbers 8 and 9 that were skipped in the second revision

Introduction

Line 68  “type I”  ->  “type 1”;  “type II -> type 2”

Line 72  “type I”  ->  “type 1”;

line 70  “I/TGFBR1” ->  “1/TGFBR1”

Line 74 and 78 “TGF-beta” -> “TGFb”

Line 88  “(MIM#614816) aortic aneurysm” ->  “(MIM#614816), unlike Marfan, aortic   aneurysm”

Line 89  “mainly the sinus of” -> “mainly, but not exclusively the sinuses of”

Line 90: eliminated: “in Marfan syndrome only the sinus of Valsalva is considered for diagnosis”

Line 90   “and a lower” -> “and has a lower”

Line 91  “LDS types was observed” ->  “LDS types”

Line 101  “in two patients with these features who”  -> in two patients who”

Line 107-109  After “mechanism [7]” ->  “Other chromosomal deletions were reported by Fontana et al , 5.2 Mb deletion [8], Gaspar et al., 4.7 Mb deletion [9], Schepers et al, 3.2 Mb deletion [10]”  has been added       

Materials and methods

Line 118 added  “Imaging analysis”

Lines 119-122

Aortic dimensions were assessed at end-diastole in the parasternal long-axis view at four levels (aortic anulus,  sinuses of Valsalvasinotubular junction and proximal ascending aorta by the leading edge-to-leading edge technique. Z-score was calculated according to appropriate age-adjusted nomograms [11,12].

Has been changed in

Aortic dimensions were assessed by trans-thoracic echocardiography at end-diastole in the parasternal long-axis view at sinuses of Valsalva, sinotubular junction, and proximal ascending aorta) by the leading edge-to-leading edge technique and Z-score was calculated according to age-adjusted nomograms [8,9].

Line 123 added “Genomic DNA preparation”

Line 127 added “Next generation sequencing (NGS) analysis“

Line 141 and  148 [10]  -> [13]

Line 151  [11]  -> [14]

Results

Lines 154-160

“The propositus is a 43-year-old male (Fig. 1, proband II-2), who at 33 was  with MFS. The patient displayed an aortic root of 47 mm (z-score 4.4), systemic features with a score of 8, hypermobility Beighton scale 8/9 (Table 2). Normal the diameter of the abdominal aorta. After 2 and 5 years respectively, aortic root diameter reached 48 and 49 mm (z-score 4.6). A mild abdominal ectasia (32mm) was detected at the age of 38. In the meanwhile a second son (we have never seen the first one born in 2013, nor the second one) was born.”

We changed this sentence as following

“The propositus is a 43-year-old male (Figure 1, proband II-2), who at 33 was diagnosed with MFS. The patient displayed an aortic root of 47 mm (z-score 4.4), with normal abdominal aortic size, systemic features with a score of 8, hypermobility Beighton scale 8/9 (Table 2). Normal the diameter of the abdominal aorta. After 2 and 5 years, aortic root diameter slowly progressed to 48 and 49 mm (z-score 4.6) respectively. A mild abdominal ectasia (32mm) was detected at the age of 38. In the meanwhile a second son was born:  both were not referred to our Centre despite our suggestion.”

Line 170  “Angio-Computed Tomography (ACT”) -> “Comtepud Tomographic angiography (CTA)”

Lines 171, 172-173   “Fig 2” -> (Figure 2)”

Table 1: in red all the adds

Line 180  “ the ACT”   ->   “computed tomographic angiography”

Lines 183-190

 “At the age of 41 the patient underwent valve sparing (according to David procedur (mother 74 and father 76 years old) are still alive in good health  The older brother  has a “potential” Marfan phenotype [2], stable over time, diagnosed at our Center [3].” 

We changed the sentence in

“At the age of 41 the patient underwent valve sparing aortic root replacement (David procedure) for 50 mm root aneurysm. Importantly, the patient reported that the father had hypertension and the mother pectus carinatum, pes planus and inguinal hernias. Echocardiography did not show cardiovascular features. The parents at present (mother 74 and father 76 years old) are alive and in good health of MFS, why eye clinical ocular manifestations were excluded  by the ophtalmologist. Both parents are still alive in good health at 74 and 76. The older brother (age 47), has a “potential” Marfan phenotype [2], stable over time, diagnosed at our Center.”

Lines 190-191 “The youngest son  was”   -> “The youngest son of the propositus (Figure 1, III-2 was”

Line 192  “Fig. 1)”  ->  “(Figure 1)”

Line 193  shock [DC = defibrillation and cardioversion]  ->  shock [DC = direct current] 

Line 197 “propositus revealed” ->  “propositus (Figure 1, III-2) revealed”

Line 200  “The older one, seen by the Meyer”  -> “The older son of the propositus (Figure 1, III-1), assessed by the Meyer”

Line 201  “Fig. 4)”  -> “(Figure 4)”

Line 202  “son at”  ->  “son (Figure 1, III-2) at”

Line 207  “son presented”  ->    “son (Figure 1, III-1)”)

Lines 213-214  “son of propositus showed a normal”  ->

“son of the propositus showed both a normal ”

Lines 224-225“sons . the brother  has”  ->  “sons (Figure 1, III-1 and III-2) The brother (Figure 1, II-3) has”

Line 230  “Our patient (Figure 1)  presented the typical manifestations of Marfan syndrome “ ->        “Our patient (Figure 1, II-2) presented the typical manifestations of MFS (Table 1)”

Line 250 “5.2 [12], 4.7 [13], 3.2 [14], 3.5”  ->  “5.2 [8], 4.7 [9], 3.2 [10], 3.5”

Line 310  “array-CGH analysis,  it would”   ->   “array-CGH analysis, this family report shows that it would”
